# Extract of *Scutellaria baicalensis* induces semaphorin 3A production in human epidermal keratinocytes

**Yasuko Yoshioka**[1,2☯], **Yayoi Kamata**[2☯], **Mitsutoshi Tominaga**[2☯], **Yoshie Umehara**[2], **Ikuyo Yoshida**[1], **Nobuya Matsuoka**[1], **Kenji Takamori**[2,3] *

**1** Central R&D Laboratory, Kobayashi Pharmaceutical Co. Ltd., Ibaraki, Osaka, Japan, **2** Juntendo Itch Research Center (JIRC), Institute for Environmental and Gender Specific Medicine, Juntendo University Graduate School of Medicine, Urayasu, Chiba, Japan, **3** Department of Dermatology, Juntendo University Urayasu Hospital, Urayasu, Chiba, Japan

☯ These authors contributed equally to this work.
* ktakamor@juntendo.ac.jp

**Data Availability Statement:** All relevant data are within the manuscript and its Supporting information files.

**Funding:** This work was supported by a grant from the Strategic Research Foundation Grant-aided

## Abstract

In a disease-state-dependent manner, the histamine-resistant itch in dry skin-based skin diseases such as atopic dermatitis (AD) and xerosis is mainly due to hyperinnervation in the epidermis. Semaphorin 3A (Sema3A) is a nerve repulsion factor expressed in keratinocytes and it suppresses nerve fiber elongation in the epidermis. Our previous studies have shown that Sema3A ointment inhibits epidermal hyperinnervation and scratching behavior and improves dermatitis scores in AD model mice. Therefore, we consider Sema3A as a key therapeutic target for improving histamine-resistant itch in AD and xerosis. This study was designed to screen a library of herbal plant extracts to discover compounds with potential to induce Sema3A in normal human epidermal keratinocytes (NHEKs) using a reporter gene assay, so that positive samples were found. Among the positive samples, only the extract of *S. baicalensis* was found to consistently increase Sema3A levels in cultured NHEKs in assays using quantitative real-time PCR and ELISA. In evaluation of reconstituted human epidermis models, the level of Sema3A protein in culture supernatants significantly increased by application of the extract of *S. baicalensis*. In addition, we investigated which components in the extract of *S. baicalensis* contributed to Sema3A induction and found that baicalin and baicalein markedly increased the relative luciferase activity, and that baicalein had higher induction activity than baicalin. Thus, these findings suggest that *S. baicalensis* extract and its compounds, baicalin and baicalein, may be promising candidates for improving histamine-resistant itch via the induction of Sema3A expression in epidermal keratinocytes.

## Introduction

Itching is a sensation of discomfort that leads to scratching. Histamine is a well-known pruritogen. Clinically, histamine $H_1$ receptor antagonists are commonly used to treat pruritus, but

Project for Private Universities from the Ministry of Education, Culture, Sports, Science and Technology (Grant number: S1311011). The funders had no role in study design, data collection and analysis, decision to publish, or preparation of the manuscript. This research was also funded by Kobayashi Pharmaceutical Co., Ltd. (URL: KOBAYASHI Pharmaceutical Co.,Ltd.). Three of the authors (YY, IY, NM), who are employees of Kobayashi Pharmaceutical Co., Ltd, had a partial role in data collection and analysis, and preparation of the manuscript.

**Competing interests:** Three of the authors (YY, IY, NM) are employees of Kobayashi Pharmaceutical Co., Ltd. We have lodged an application for a domestic patent. This application has not been approved, and is currently pending. This does not alter our adherence to PLOS ONE policies on sharing data and materials.

they are often ineffective against pruritic skin diseases, such as atopic dermatitis (AD) and xerosis [1, 2].

Histologically, cutaneous sensory nerve fibers are localized in the vicinity of the boundary between the epidermis and the dermis in normal skin. On the other hand, in the AD or xerosis skin in the disease-state, a large numbers of sensory nerve fibers penetrated and sprouted through the basal cell layer, resulting in epidermal hyperinnervation. The nerve fibers extending just under the stratum corneum with the skin barrier disruption are thought to cause itch hypersensitivity against external stimuli [1–6].

Our previous studies demonstrated that epidermal innervation is regulated by the balance between nerve elongation factors (*e.g.* nerve growth factor [NGF]) and nerve repulsion factors (*e.g.* semaphorin 3A [Sema3A]) [4]. NGF produced by keratinocytes is one of the major growth factors that determines skin innervations, with higher local NGF concentrations in the lesional skin of patients with AD or xerosis than in normal skin [1]. Decreased levels of Sema3A expression in the epidermis have been reported in patients with AD [7] and in an experimental model [8], concomitant with an increase in epidermal nerve density. Sema3A has been shown to inhibit NGF-induced sprouting of sensory afferents in the adult mammalian spinal cord [9], and elevated levels of NGF reduced the Sema3A-induced collapse of sensory growth cones [10]. These findings suggest that decreasing Sema3A levels in the epidermis can accelerate epidermal nerve growth in dry-skin-based diseases.

Experimentally, it has been reported that replacement of recombinant Sema3A in the lesional skin of NC/Nga mice normalized epidermal hyperinnervation, resulting in suppression of itch-related scratching behavior and improved dermatitis [11, 12]. This raises the possibility that enhancement of epidermal Sema3A levels may be a promising therapeutic approach for improving pruritus associated with intra-epidermal nerve density. It is generally thought that small molecules (<500 Da) pass through the normal skin barrier [13], whereas larger molecules, such as recombinant Sema3A with a molecular weight of 113.5 kDa, would be less likely to penetrate the stratum corneum barrier. Even if it is absorbed into the epidermis from the cracked stratum corneum of lesional skin, a protein molecule also risks side effects such as contact dermatitis. In addition, recombinant Sema3A is an unstable substance in ointment base [11]. It is likely that there may be many problems in the clinical application of Sema3A direct replacement therapy. Therefore, we considered that topical application of a low-molecular-weight compound with the potential to induce endogenous Sema3A production from human epidermal keratinocytes, rather than Sema3A itself, is a useful approach to improve itch caused by epidermal hyperinnervation.

In this study, we screened a library of herbal plant extracts, which have been confirmed to be safe or use in external preparations, to discover compounds with potential to induce endogenous Sema3A production from normal human epidermal keratinocytes (NHEKs). Here, we describe effect of *Scutellaria baicalensis* extract on Sema3A production in human epidermal keratinocytes.

## Materials and methods

### Preparation of test samples

The ninety-one herbal plants used for screening in this study are shown in Table 1. Ethanol extracts of the herbal plants were obtained from Maruzen Pharmaceuticals Co., Ltd. (Hiroshima, Japan) and Koei Kogyo Co., Ltd. (Tokyo, Japan). These ethanol extracts were evaporated under reduced pressure and lyophilized as test samples. Lyophilized samples were reconstituted at 100 mg/mL with DMSO, and each test sample was diluted with basal medium at a concentration in the range of 0.02–500 μg/mL and used in each experiment.

**Table 1. A list of herbal plants used in screening of Sema3A inducers.**

| No | Scientific name | Part | No. | Scientific name | Part |
|---|---|---|---|---|---|
| 1 | *Uncaria gambir* | Leaf, burgeon | 47 | *Ziziphus jujuba* | Fruit |
| 2 | *Arnica montana* | Flower | 48 | *Thymus vulgaris* | Aerial part |
| 3 | *Aloe ferox, Aloe africana* | Leaf | 49 | *Syzygium aromaticum* | Bud |
| 4 | *Aloe arborescens* | Leaf | 50 | *Citrus unshiu* | Pericarp |
| 5 | *Ginkgo biloba* | Leaf | 51 | *Capsicum annuum* | Fruit |
| 6 | *Urtica thunbergiana* | Leaf | 52 | *Angelica acutiloba* | Root |
| 7 | *Artemisia capillaris* | Flower head | 53 | *Calendula officinalis* | Flower |
| 8 | *Foeniculum vulgare* | Fruit | 54 | *Panax ginseng* | Root |
| 9 | *Thea sinensis* | Leaf | 55 | *Lonicera japonica* | Leaf, stem |
| 10 | *Malva sylvestris L.* | Flower | 56 | *Eriobotrya japonica* | Leaf |
| 11 | *Rosa multiflora* | Fruit | 57 | *Tussilago farfata* | Flower, leaf |
| 12 | *Rabdosia japonica* | Aerial part | 58 | *Poria cocos* | Sclerotium |
| 13 | *Scutellaria baicalensis* | Root | 59 | *Ruscus aculeatus* | Rootstock |
| 14 | *Phellodendron amurense* | Bark | 60 | *Carthamus tinctorius* | Flower |
| 15 | *Fucus vesiculosus* | Whole alga | 61 | *Mentha piperita* | Leaf |
| 16 | *Glycyrrhiza glabra, Glycyrrhiza uralensis* | Root, rootstock | 62 | *Paeonia suffruticosa* | Root bark |
| 17 | *Cinchona succirubra* | Bark | 63 | *Humulus lupulus* | Female Flower |
| 18 | *Prunus armeniaca* | Seed | 64 | *Aesculus hippocastanum* | Seed |
| 19 | *Mallotus philippinensis* | Bark | 65 | *Melissa officinalis* | Leaf |
| 20 | *Sasa veitchii* | Leaf | 66 | *Prunus persica* | Leaf |
| 21 | *Matricaria chamomilla* | Flower | 67 | *Eucalyptus globulus* | Leaf |
| 22 | *Sophora flavescens* | Root | 68 | *Saxifraga stolonifera* | Whole plant |
| 23 | *Citrus paradisi* | Fruit | 69 | *Citrus junos* | Fruit |
| 24 | *Cinnamomum cassia* | Bark | 70 | *Lavandula angustifolia* | Flower |
| 25 | *Gentiana lutea* | Root, rootstock | 71 | *Thea sinensis* | Leaf |
| 26 | *Geranium thunbergii* | Aerial part | 72 | *Ganoderma lucidum* | Fruit body |
| 27 | *Saccharomyces cerevisiae* | Whole | 73 | *Rosa canina* | Fruit |
| 28 | *Arctium lappa* | Root | 74 | *Rosmarinus officinalis* | Leaf |
| 29 | *Oryza sativa* | Seed coat | 75 | *Thymus serpyllum* | Aerial part |
| 30 | *Asiasarum sieboldii* | Root, rootstock | 76 | *Hydrangea serrata var. thunbergii* | Leaf |
| 31 | *Salvia officinalis* | Leaf | 77 | *Coptis japonica* | Rootstock |
| 32 | *Crataegus cuneata* | Fruit | 78 | *Hypericum perforatum* | Aerial part |
| 33 | *Gardenia jasminoides* | Fruit | 79 | *Artemisia princeps* | Leaf |
| 34 | *Zanthoxylum piperitum* | Pericarp | 80 | *Lonicera japonica* | Flower |
| 35 | *Rehmannia glutinosa* | Root | 81 | *Thea sinensis var. assamica* | Leaf |
| 36 | *Perilla frutescens var. Crispa* | Leaf | 82 | *Acorus calamus* | Rootstock |
| 37 | *Tilia cordata* | Flower | 83 | *Achillea millefolium* | Whole plant |
| 38 | *Paeonia lactiflora* | Root | 84 | *Morus alba* | Root bark |
| 39 | *Houttuynia cordata* | Aerial part | 85 | *Luffa cylindrica* | Aerial part |
| 40 | *Sanguisorba officinalis* | Root, rootstock | 86 | *Lilium candidum* | Bulb |
| 41 | *Zingiber officinale* | Rootstock | 87 | *Coix lachryma-jobi var. ma-yuen stapf* | Seed |
| 42 | *Betula pendula* | Bark | 88 | *Cordyceps sinensis* | Whole |
| 43 | *Pinctada fucate* | Nacre | 89 | *Machilus odoratissima* | Bark |
| 44 | *Equisetum arvense* | Whole | 90 | Punica granatum | Pericarp |
| 45 | *Cnidium officinale* | Rootstock | 91 | *Vitis vinifera* | Leaf |
| 46 | *Swertia japonica* | Whole plant | | | |

## Cell culture

NHEKs derived from adult epidermis (Lonza, Basel, Switzerland) were cultured in keratinocyte basal medium (KBM-Gold; Lonza, Basel, Switzerland) supplemented with KGM-Gold SingleQuot (Lonza), containing 0.15 mM calcium, at 37˚C with 5% $CO_2$. Cells were passaged at 60–70% confluence, and the experiments were performed using sub-confluent cells within four passages.

## Transfection and measurement of promoter activity

NHEKs were seeded in 24-well flat bottom plates for 24 hours. The culture medium was replaced with antibiotic-free medium and the cells were co-transfected with 500 ng/well pGL3-*Sema3A* vector containing the *firefly* luciferase gene and 50 ng/well pRL3-TK vector containing the *Renilla* luciferase gene (Promega, Madison, WI, USA) using X-tremeGENE HP DNA Transfection Reagent (Roche Diagnostics GmbH, Mannheim, Germany) in accordance with the manufacturer's instructions. pGL3-*Sema3A* vector was produced in accordance with the methods reported by Kamata et al. [14]. At 24 hours after the start of transfection, the medium was replaced with fresh medium containing the test samples shown in Table 1, apigenin, baicalein, baicalin, chrysin, or wogonin (Wako Pure Chemical Industries, Ltd., Osaka, Japan) at various concentrations, and NHEKs were further incubated for 24 or 48 hours at 37˚C. The cells were washed and lysed with passive lysis buffer, and luciferase activities were analyzed using a dual-luciferase reporter assay system with a 2030 ARVO-X4 Multi label Plate Reader (Perkin Elmer, Waltham, MA, USA) in accordance with the manufacturer's instructions. Relative luciferase activity was expressed as a relative ratio of experimental data to untreated control data after normalization with *firefly* luminescence/*Renilla* luminescence ratio for each sample well. Cultured cells were observed using a microscopic and checked for morphologically damaged cells by test sample treatment. Data obtained from damaged cells were excluded from the results.

## Measurement of Sema3A at mRNA and protein levels in cultured NHEKs

NHEKs were incubated with five plant extracts, *Arnica montana* (sample No. 2), *Artemisia capillaris* (No. 7), *Malva sylvestris L.* (No. 10), *Scutellaria baicalensis* (No. 13), or *Hydrangea serrata var. thunbergii* (No. 76), at a final concentration of 50 μg/mL for 3 or 6 hours at 37˚C. The cells were subsequently harvested and transcription levels of the human *Sema3A* gene were quantified using quantitative real-time PCR. Quantification was performed using an Applied Biosystems Fast Real-Time PCR system (Applied Biosystems) using PrimeScript™ One Step RT-PCR Kit (Takara Bio Inc., Shiga, Japan) and the primers 5′–ACCCAACTATCAA TGGGTGCCTTA–3′ (forward) and 5′–AACACTGGATTGTACATGGCTGGA–3′ (reverse). As an internal control, *ribosome protein S18* (*RPS18*) mRNA was amplified using the primers 5′–TTTGCGAGTACTCAACACC AACATC–3′ (forward) and 5′–GAGCATA TCTTCGGCCCA CAC–3′ (reverse). The amounts of mRNA were normalized relative to those of *RPS18* and expressed relative to ratios of the untreated controls.

The culture supernatants were collected after 3, 6, or 24 hours and Sema3A protein concentration was measured using an enzyme-linked immunosorbent assay (ELISA) kit for Sema3A (Uscn Life Science, Wuhan, China), in accordance with the manufacturer's protocol. Cells cultured for 6 or 24 hours were lysed in M-PER mammalian protein extraction reagent and protease inhibitor cocktail (Thermo Scientific, Waltham MA, USA), then Sema3A levels in the cell lysate were measured using a Human Sema3A ELISA kit (Elabscience, Houston, TX, USA).

## Cell counting assay

A Cell Counting Kit (CCK)- 8 (Dojindo, Kumamoto, Japan) was used to assess the cell toxicity of each compound on NHEKs. Cells were seeded at $1 \times 10^4$ cells per well in 96-well culture plates. After incubation overnight at 37˚C, an extract of *S. baicalensis* or *A. montana* (50 μg/mL) was added to each well, then incubated for 6 or 24 hours at 37˚C. At each time point, CCK-8 solution was added to each well and incubated for 3 hours at 37˚C. Cell viability was estimated by measuring the absorbance at 450 nm using a 2030 ARVO-X4 Multi label Plate Reader (Perkin Elmer).

## Reconstructed human epidermis (RHE) models

RHE model (EPISKIN, SA, Lyon, France) is an *in vitro* reconstructed human epidermis from normal human keratinocytes cultured on an inert polycarbonate filter at the air-liquid inter-face, in a chemically defined medium, featuring normal ultrastructure and functionality simi-lar to human tissue *in vivo* [15]. RHE models were cultured in accordance with the manufacturer's protocol. Briefly, inserts containing the RHE models were shipped at room temperature in a multi-well plate filled with an agarose-nutrient solution in which they were embedded. The inserts were carefully taken out of the multi-well plate, any remaining agarose adhering to the outer sides of the insert was removed, and the inserts were placed in a plate in which each well has been filled with Maintenance Medium (EPISKIN, Lyon, France) overnight at 37˚C, 5% $CO_2$. Next, 100 μL of the extract of *A. montana* or *S. baicalensis* (100 or 500 μg/mL) was applied from the stratum corneum side of the reconstructed epidermis samples and cultured for 24 hours. The concentration of Sema3A protein in the culture supernatants was measured by ELISA as described above.

## Statistical analyses

Unless otherwise indicated, values were presented as the standard deviation (S.D.) of the mean. Statistical analyses were performed using one-way or two-way ANOVA followed by Dunnett's multiple comparison tests with GraphPad Prism 8 (GraphPad Software, Inc., CA, USA), with $p < 0.05$ defined as statistically significant.

# Results

## Screening of Sema3A inducers from herbal plant extracts using a reporter gene assay

Cultured NHEKs were transfected with luciferase reporter gene constructs containing -1444 bp of the human *Sema3A* promoter as described previously [14]. Following transfection, cells were treated with various types of herbal plant extracts at a final concentration of 50 μg/mL for 24 hours and the cell lysates were subjected to luciferase assays. The results are shown in Table 2. From 91 different types of herbal plant extracts, 48 extracts exhibited significantly increased relative luciferase activity compared with the untreated controls, with no cell damage morphologically by the addition thereof. The dose–response relationship was examined with the lower concentration range for the top 5 samples (sample Nos. 2, 7, 10, 13, and 76) among the 48 herbal plants extracts showing significantly higher activity compared with the control. The samples of No. 7, 10, and 76 increased activity in a dose-dependent manner (Fig 1). No dose-dependent increases in activity were observed with samples No. 2 and 13, although in sample No. 13, it seemed that its activity probably reached a plateau, because it was the most active at a low concentration of 5 μg/mL.

**Table 2. A summary of *Sema3A* reporter gene activity by herbal plant extracts.**

| No. | Mean | S.D. | Significant difference | P value | No. | Mean | S.D. | Significant difference | P value |
|---|---|---|---|---|---|---|---|---|---|
| 1 | 1.60 | 0.07 | *** | 0.0007 | 47 | 1.05 | 0.06 | n.s. | 0.9994 |
| 2 | 2.21 | 0.20 | **** | < 0.0001 | 48 | 1.24 | 0.14 | n.s. | 0.367 |
| 3 | 1.62 | 0.18 | *** | 0.000 | 49 | 1.43 | 0.12 | * | 0.0108 |
| 4 | 1.53 | 0.07 | ** | 0.004 | 50 | 1.02 | 0.01 | n.s. | 0.9998 |
| 5 | 1.60 | 0.06 | *** | 0.001 | 51 | 1.62 | 0.10 | *** | 0.0002 |
| 6 | 1.67 | 0.13 | *** | 0.000 | 52 | 1.08 | 0.05 | n.s. | 0.999 |
| 7 | 2.81 | 0.21 | **** | < 0.0001 | 53 | 1.29 | 0.27 | n.s. | 0.1762 |
| 8 | 1.73 | 0.07 | **** | < 0.0001 | 54 | 1.23 | 0.09 | n.s. | 0.3833 |
| 9 | 1.55 | 0.30 | ** | 0.002 | 55 | 1.33 | 0.03 | n.s. | 0.0841 |
| 10 | 1.97 | 0.16 | **** | < 0.0001 | 56 | 1.38 | 0.30 | * | 0.0348 |
| 11 | 1.44 | 0.17 | * | 0.02 | 57 | 1.41 | 0.13 | * | 0.019 |
| 12 | 1.39 | 0.17 | n.s. | 0.05 | 58 | 1.29 | 0.12 | n.s. | 0.996 |
| 13 | 1.95 | 0.09 | **** | < 0.0001 | 59 | 1.49 | 0.14 | n.s. | 0.9042 |
| 14 | 1.03 | 0.06 | n.s. | 1.00 | 60 | 1.38 | 0.08 | n.s. | 0.9829 |
| 15 | 1.32 | 0.06 | n.s. | 0.20 | 61 | 1.40 | 0.21 | n.s. | 0.9741 |
| 16 | 1.69 | 0.16 | **** | < 0.0001 | 62 | 1.55 | 0.14 | n.s. | 0.8322 |
| 17 | 1.29 | 0.23 | n.s. | 0.31 | 63 | 0.67 | 0.05 | n.s. | 0.992 |
| 18 | 1.19 | 0.06 | n.s. | 0.81 | 64 | No data | No data | - | - |
| 19 | No data | No data | - | - | 65 | 1.62 | 0.10 | n.s. | 0.7277 |
| 20 | 1.40 | 0.07 | * | 0.05 | 66 | 1.25 | 0.13 | n.s. | 0.9991 |
| 21 | 1.25 | 0.03 | **** | < 0.0001 | 67 | 1.57 | 0.18 | n.s. | 0.7978 |
| 22 | 1.20 | 0.03 | **** | < 0.0001 | 68 | 1.03 | 0.15 | n.s. | 0.9999 |
| 23 | 1.03 | 0.02 | n.s. | 0.8836 | 69 | 1.13 | 0.04 | n.s. | 0.9996 |
| 24 | 1.24 | 0.05 | **** | < 0.0001 | 70 | 1.36 | 0.26 | n.s. | 0.9893 |
| 25 | 1.19 | 0.04 | **** | < 0.0001 | 71 | 1.31 | 0.11 | ** | 0.0062 |
| 26 | 1.25 | 0.05 | **** | < 0.0001 | 72 | 1.35 | 0.01 | *** | 0.0006 |
| 27 | 1.20 | 0.02 | **** | < 0.0001 | 73 | 1.16 | 0.04 | n.s. | 0.2914 |
| 28 | 1.23 | 0.01 | **** | < 0.0001 | 74 | 1.38 | 0.07 | *** | 0.0006 |
| 29 | 1.19 | 0.02 | **** | < 0.0001 | 75 | 1.42 | 0.13 | **** | < 0.0001 |
| 30 | 1.23 | 0.02 | **** | < 0.0001 | 76 | 2.27 | 0.17 | **** | < 0.0001 |
| 31 | 1.34 | 0.02 | **** | < 0.0001 | 77 | 1.29 | 0.03 | ** | 0.0062 |
| 32 | 1.27 | 0.00 | **** | < 0.0001 | 78 | 1.38 | 0.06 | *** | 0.0006 |
| 33 | 1.28 | 0.05 | **** | < 0.0001 | 79 | 1.54 | 0.08 | **** | < 0.0001 |
| 34 | 1.30 | 0.01 | **** | < 0.0001 | 80 | 1.33 | 0.04 | *** | 0.0006 |
| 35 | 0.96 | 0.26 | n.s. | 0.9993 | 81 | 1.32 | 0.02 | ** | 0.0062 |
| 36 | 1.17 | 0.03 | n.s. | 0.2271 | 82 | 1.18 | 0.09 | n.s. | 0.2914 |
| 37 | 1.10 | 0.03 | n.s. | 0.7929 | 83 | 1.38 | 0.13 | *** | 0.0002 |
| 38 | 1.19 | 0.03 | n.s. | 0.1783 | 84 | 0.02 | 0.01 | **** | < 0.0001 |
| 39 | 1.18 | 0.03 | n.s. | 0.2098 | 85 | 0.99 | 0.08 | n.s. | 0.9997 |
| 40 | 1.26 | 0.03 | * | 0.0244 | 86 | 0.96 | 0.07 | n.s. | 0.9534 |
| 41 | 1.26 | 0.24 | * | 0.0197 | 87 | 0.99 | 0.08 | n.s. | 0.9998 |
| 42 | 1.10 | 0.01 | n.s. | 0.82 | 88 | 0.87 | 0.04 | n.s. | 0.8227 |
| 43 | 1.24 | 0.01 | * | 0.0368 | 89 | 1.01 | 0.22 | n.s. | > 0.9999 |
| 44 | 1.31 | 0.01 | ** | 0.0037 | 90 | 0.85 | 0.13 | n.s. | 0.398 |
| 45 | 1.19 | 0.01 | n.s. | 0.1383 | 91 | 0.83 | 0.04 | n.s. | 0.3129 |

(*Continued*)

**Table 2.** (Continued)

| No. | Mean | S.D. | Significant difference | P value | No. | Mean | S.D. | Significant difference | P value |
|---|---|---|---|---|---|---|---|---|---|
| 46 | 1.27 | 0.07 | * | 0.0143 | | | | | |

NHEKs were transfected with pGL3-*Sema3A*. After transfection, the cells were incubated with the herbal plant extracts shown in Table 1 at a final concentration of 50 μg/mL for 24 hours at 37°C. Cell lysates were subjected to luciferase activity measurement as described in the Materials and methods. All results were expressed as mean ± S.D. of triplicate samples.

* $p < 0.05$,

** $p < 0.01$,

*** $p < 0.001$,

**** $p < 0.0001$, n.s: no significant difference (one-way ANOVA followed by Dunnett's test). No data: data obtained from cells in which morphological damage was observed by the addition of sample was not used.

## Effects of the herbal plant extracts Sema3A induction at mRNA and protein levels in NHEKs

We next examined the induction of Sema3A mRNA and protein levels with the five positive plants extracts (sample No. 2, 7, 10, 13, and 76) by quantitative real-time PCR and ELISA, respectively. Expression of *Sema3A* mRNA in the NHEKs treated with No. 2, 7, and 13 was markedly increased at both 3 and 6 hours compared with untreated controls (Fig 2). In samples No. 10 and 76, they were slightly increased at 3 and 6 hours, respectively.

Sema3A protein levels were measured in the culture supernatants of NHEKs treated with five tested samples using ELISA (Fig 3A). At 3 hours after sample treatment, differences in Sema3A protein levels were hardly noted in the culture supernatants compared with the control group. However, from 6 hours onward, sample treatment led to an overall increase in Sema3A expression, and Sema3A levels were significantly increased with sample No. 13 compared with the controls at 24 hours. In sample No. 2, we noted a tendency to increase, although no statistically significant difference was observed. Increased expression at the protein level was also observed using ELISA with cell lysates of cultured NHEKs with sample No. 13 but not sample No. 2 (Fig 3B). We further tested the effects of samples No. 2 and No. 13 on cell

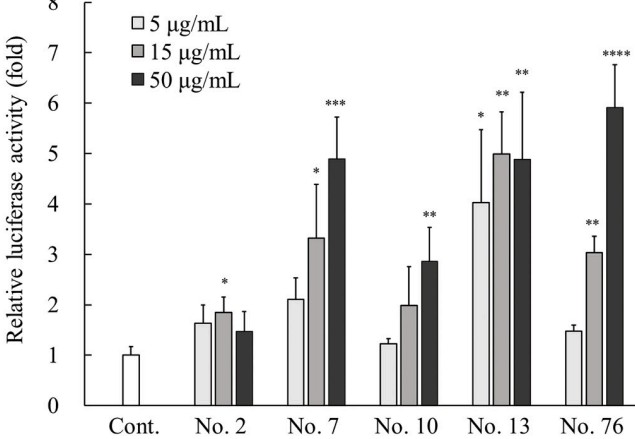

**Fig 1. Effects of five positive hits of plant extracts on Sema3A reporter gene expression.** NHEKs were transfected with pGL3-*Sema3A*. After transfection, the cells were incubated with five plant extracts, namely No. 2, 7, 10, 13, and 76 at a final concentration of 5, 15, and 50 μg/mL for 48 hours at 37°C. Cell lysates were subjected to luciferase activity measurement as described in Materials and methods. All results are expressed as mean ± SD of three independent experiments. * $p < 0.05$, ** $p < 0.01$, *** $p < 0.001$, **** $p < 0.0001$ (one-way ANOVA followed by Dunnett's test).

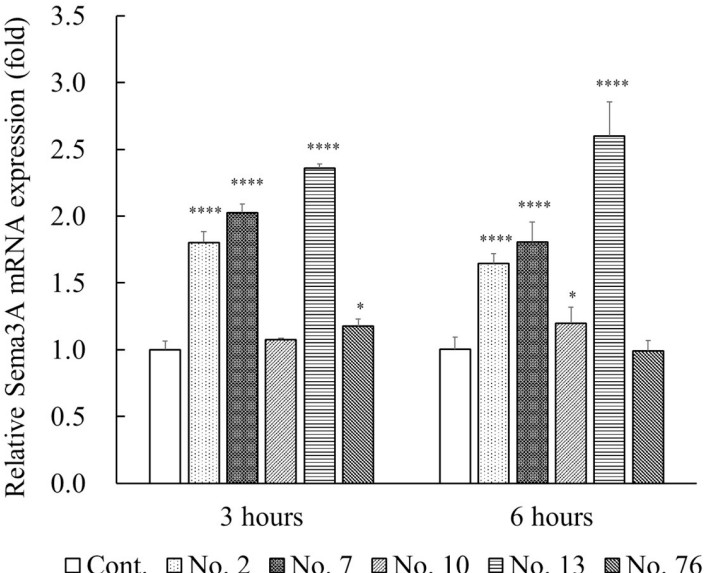

**Fig 2. Effects of five positive plant extracts on the expression of *Sema3A* mRNA in cultured NHEKs.** NHEKs were incubated with five plant extracts, samples No. 2, 7, 10, 13, and 76 at a final concentration of 50 μg/mL for 3 and 6 hours at 37˚C. Expression of *Sema3A* mRNA was examined using quantitative real-time PCR analysis. The levels of *Sema3A* mRNA were normalized relative to those of *RPS18*. All results are expressed as mean ± S.D. of three independent experiments. $^{*}$ $p < 0.05$, $^{**}$ $p < 0.01$, $^{***}$ $p < 0.001$, $^{****}$ $p < 0.0001$ (one-way ANOVA followed by Dunnett's test).

viability of cultured NHEKs using a cell counting assay. In the assay, the cell viabilities of NHEKs treated with these compounds was significantly increased at 6 hours compared with untreated controls, whereas samples No. 2 and No. 13 reduced cell viabilities to 89.8% and 78.2% at 24 hours, respectively (Fig 3C).

## Effect of the plant extracts on Sema3A protein production in RHE models

The Sema3A-inducing activity of the extracts of *S. baicalensis* and *A. montana* was evaluated using RHE models. These plant extracts at concentrations of 100 or 500 μg/mL were applied to the stratum corneum side of RHE models. They were cultured for 24 hours, and then Sema3A protein levels in the culture supernatant were measured. In the *S. baicalensis*-treated group, the concentration of Sema3A increased in a dose-dependent manner, with significantly higher values noted in the 500 μg/mL *S. baicalensis*-treated group compared with the untreated control (Fig 4). Meanwhile, application of *A. montana* extract did not affect Sema3A concentration in the culture supernatant.

## Bioactive components in *S. baicalensis* contributing to Sema3A induction

We examined which components in the extract of *S. baicalensis* contributed to Sema3A induction (Fig 5A). Of the known chemical components in the extract of *S. baicalensis*, we conducted *Sema3A* reporter gene assays in NHEKs treated with the five major bioactive flavones, baicalin, baicalein, wogonin, chrysin, and apigenin. The experiment was performed with a maximum concentration of 5 μg/mL and a 3-fold dilution series. The molar concentration of each compound is shown in S1 Table. The extract of *S. baicalensis*, baicalin, and baicalein all significantly increased the relative luciferase activity at a concentration of 5 μg/mL, and the increase in activity of these samples was dose dependent. Notably, baicalein markedly

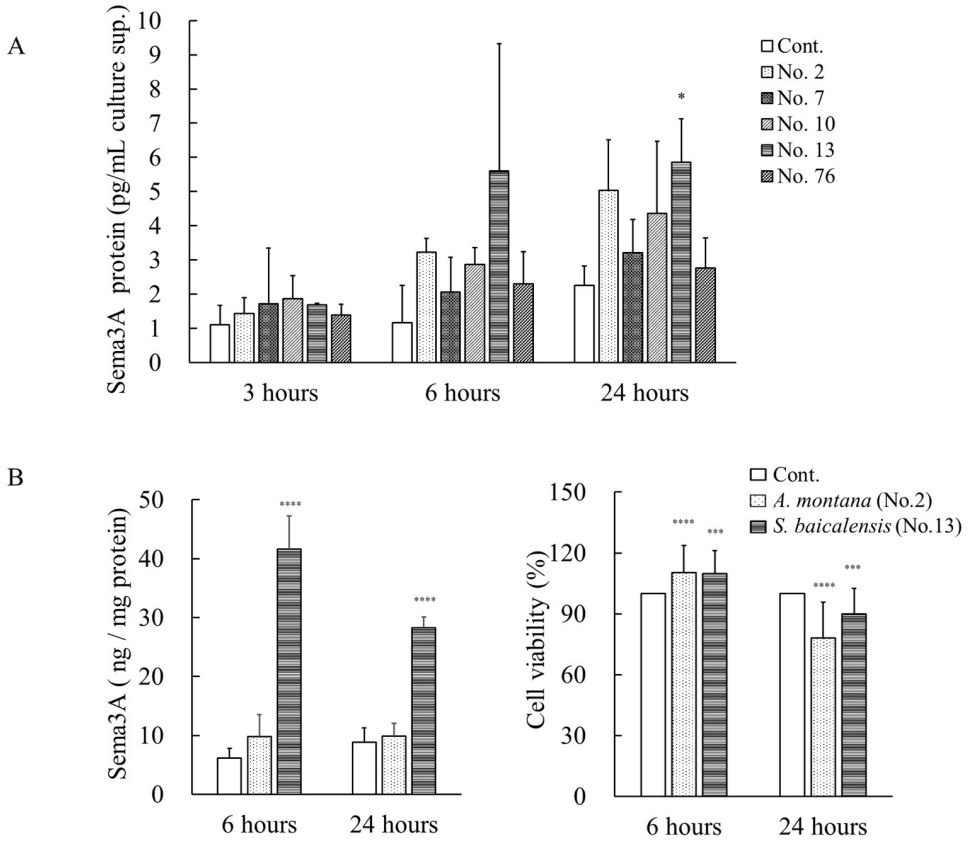

**Fig 3. Effects of five positive plant extracts on the production of Sema3A proteins in cultured NHEKs.** (A) NHEKs were incubated with five plant extracts, samples No. 2, 7, 10, 13, and 76 at a final concentration of 50 μg/mL for 3, 6, and 24 hours at 37°C. The culture supernatants were collected, and then the concentrations of Sema3A protein were measured using ELISA. *$p < 0.05$, (one-way ANOVA followed by Dunnett's test). (B) NHEKs were incubated with sample No. 2 (*A. montana*) or No. 13 (*S. baicalensis*) at 50 μg/mL for 6 and 24 hours. After that, the cell lysates were prepared with lysis buffer, and then the levels of Sema3A protein in the cell lysates were measured using ELISA. (C) Cell viability of cultured NHEKs treated with the extract of *A. montana* or *S. baicalensis* was measured using a cell counting assay. All results are expressed as means ± S.D. of three independent experiments. * $p < 0.05$, ** $p < 0.01$, *** $p < 0.001$, **** $p < 0.0001$ (two-way ANOVA followed by Dunnett's test).

increased its activity at a lower concentration of 0.56 μg/mL compared with the extract of S. *baicalensis* and baicalin. Such significant alterations were observed with chrysin and apigenin but were not dose dependent. Meanwhile, wogonin significantly decreased the luciferase activity at a concentration of 5 μg/mL

## Discussion

In the present study, we found five positive samples from 91 herbal plant extracts as potent inducers for human *Sema3A* gene in NHEKs using our *Sema3A* reporter gene assay. Among these, samples No. 2, 7, and 13 markedly increased the expression levels of *Sema3A* mRNA in cultured NHEKs at 3 and 6 hours (Fig 2). In addition, the extract of *S. baicalensis* significantly increased the level of Sema3A protein in the supernatants of NHEKs cultured for 24 hours (Fig 3A) and in cell lysates from NHEKs cultured for 6 and 24 hours (Fig 3B). Meanwhile, no significant increases in the level of Sema3A protein in culture supernatants and cell lysates from NHEKs treated with the extract of *A. montana* were found in this study. These findings suggested that the extract of *S. baicalensis* consistently has Sema3A-inducing activity in

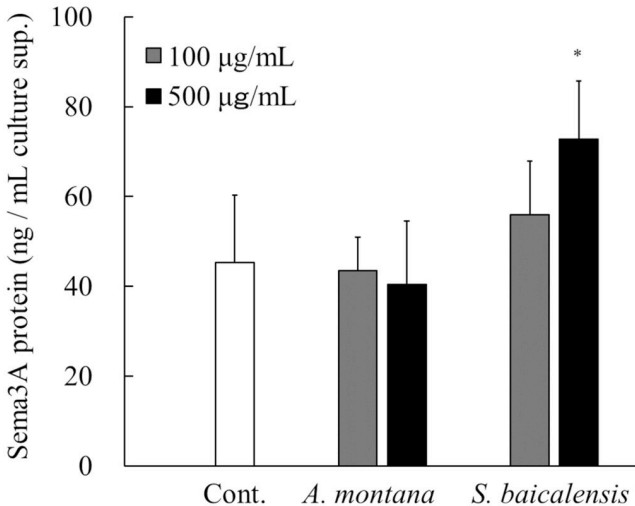

**Fig 4. Effects of *S. baicalensis* and *A. montana* extracts on Sema3A production in RHE models.** A 100 μL sample of the extract of *S. baicalensis*, *A. montana* (100 or 500 μg/mL) or PBS (vehicle) was applied from the stratum corneum side of cultured RHE samples, and the RHE samples were cultured for a further 24 hours. The culture supernatants were collected, and then the concentrations of Sema3A protein were measured using ELISA. All results are expressed as means ± S.D. of three independent experiments. $^{*}p < 0.05$, (one-way ANOVA followed by Dunnett's test).

cultured NHEKs on both gene and protein expression levels. In addition, our cell counting assay showed that extracts of *S. baicalensis* and *A. montana* significantly increased cell viability at 6 hours, whereas the cell viabilities with *S. baicalensis* and *A. montana* treatments at 24 hours reduced to 89.8% and 78.2%, respectively (Fig 3C), although no morphological findings related to cytotoxicity were observed in microscopic observations up to 24 hours. Thus, these findings raise the possibility that extracts of *S. baicalensis* and *A. montana* are at least partly involved in cell proliferation and/or differentiation rather than cytotoxic events. This idea might also be supported by a previous study that baicalein, one of the components in *S. baicalensis*, induced keratinocyte differentiation [16].

Notably, Sema3A concentration in the culture supernatants was significantly increased in the *S. baicalensis* extract-treated group in RHE model, whereas no increase was observed in the *A. montana* extract-treated group (Fig 4). This model was reconstructed by culturing human epidermal keratinocytes, featuring normal ultrastructure and functionality similar to human tissue *in vivo* [15]. Each sample was applied to the stratum corneum side of the three-dimensional skin model. It is generally thought that small molecules (<500 Da) pass through the normal skin barrier [13]. The result of RHE models indicated that at least some of the components that contribute to the induction of Sema3A in the extract from *S. baicalensis* can pass through the stratum corneum and act on keratinocytes. From these findings, it was suggested that topical application of the *S. baicalensis* extract can alleviate itch caused by hyperinnervation through induction of endogenous Sema3A in epidermal keratinocytes. In fact, our preliminary study using a pruritic dry skin model mouse also showed that repeated application of *S. baicalensis* extract tended to relieve the itch-related scratching behavior and epidermal hyperinnervation (Yoshioka et al. unpublished observations).

*S. baicalensis* has long been used as the medicinal plant in Eastern traditional herbal formulations. More than 40 kinds of flavonoids, including baicalin, baicalein, wogonin, chrysin, and apigenin, have been identified from the extract of *S. baicalensis*, which has various pharmacological activities including anti-inflammatory, anti-allergic, anti-oxidant, anti-tumor, anti-bacterial, and neuroprotective actions [17–19]. We investigated which components in the extract

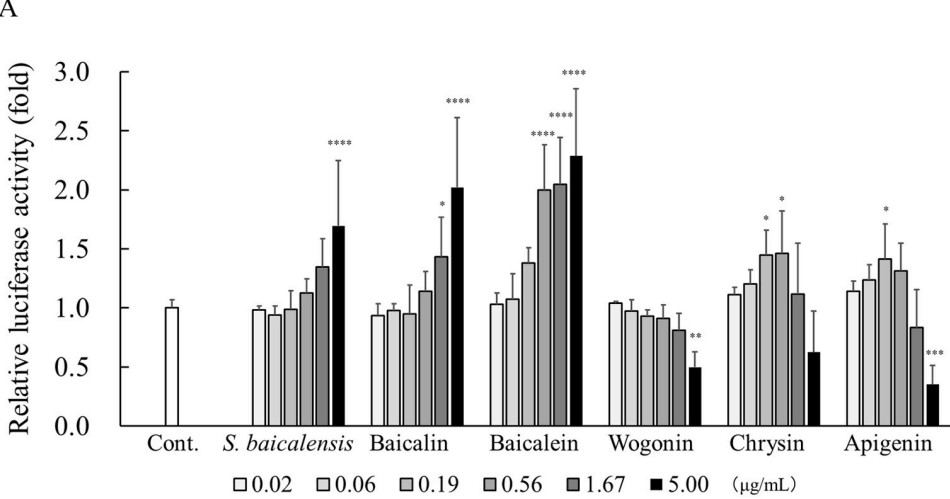

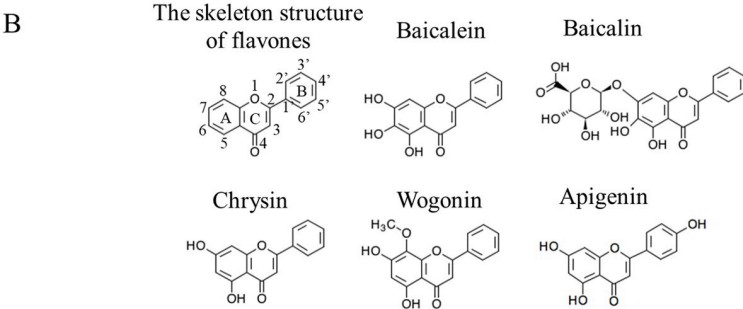

**Fig 5. Effects of *S. baicalensis*–derived compounds on *Sema3A* reporter gene expression.** (A) NHEKs were transfected with pGL3-*Sema3A* vector. After transfection, the cells were incubated with *S. baicalensis* extract and *S. baicalensis*-derived compounds: apigenin, baicalein, baicalin, chrysin, and wogonin, at the indicated concentrations, a 3-fold dilution with a maximum concentration of 5 μg/mL, for 24 hours at 37°C. Cell lysates were subjected to the measurement of luciferase activities as described in the Materials and methods. All results are expressed as means ± S.D. of three independent experiments. $^*$ $p < 0.05$, $^{**}$ $p < 0.01$, $^{***}$ $p < 0.001$, $^{****}$ $p < 0.0001$ (one-way ANOVA followed by Dunnett's test). (B) Chemical formulae of major bioactive flavones from *S. baicalensis*.

of *S. baicalensis* contributed to Sema3A induction. Among the major bioactive flavones in *S. baicalensis* subjected to the assay, baicalin, baicalein, and wogonin are components specific to *S. baicalensis* [17, 20, 21], whereas chrysin and apigenin are present in a variety of plants [22, 23]. Our reporter gene assay showed that relative luciferase activities in cultured NHEKs were significantly increased in a dose-dependent manner by the application of an extract of *S. baicalensis*, baicalin and baicalein compared with the control (Fig 5A). Notably, baicalein showed significantly higher activity at a low concentration (0.56 μg/mL) among the major bioactive flavones in *S. baicalensis*, suggesting that it is the most active compound. Baicalein is a glucuronide conjugate (Fig 5B) and is known to be converted to its aglycone, baicalein, by β-glucuronidase derived from bacteria but not that from humans [24]. It has been previously reported that an extract of *S. baicalensis* contains approximately 10% baicalin and 1% baicalein [25–27], so baicalin is a major component of *S. baicalensis* extract. Taken together, these findings suggested that baicalin and its aglycone, baicalein, are the main active components in *S. baicalensis* contributing to Sema3A induction in keratinocytes.

Among the four flavone aglycones tested, baicalein showed significantly higher activity in a dose-dependent manner, but no other aglycones, wogonin, chrysin, and apigenin. Baicalein

(5,6,7-trihydroxyflavone), chrysin (5,7-dihydroxyflavone), and wogonin (5,7-dihydroxy-8-methoxyflavone) have substituents in different patterns only in ring A (Fig 5B). Thus, the different structure in ring A might affect activity on the Sema3A promoter in epidermal keratinocytes, although the precise mechanisms remain unclear in this study. Recently, we reported that *Sema3A* mRNA expression in NHEKs is regulated by calcium via the mitogen-activated protein kinase and activator protein (AP)-1 signaling axis [14]. Meanwhile, baicalein has been shown to suppress the transcription activity of AP-1 [28]. Therefore, the extract of *S. baicalensis* may promote Sema3A production via different signaling pathways.

In conclusion, the results of present study showed that the extract of *S. baicalensis* and its active compounds, baicalin and baicalein, induced Sema3A production in normal human epidermal keratinocytes. Thus, an *S. baicalensis* extract or its active compounds may be promising therapeutic candidates for improving histamine-resistant itch caused by epidermal hyperinnervation.

## Supporting information

**S1 Table. List of concentration units in μg/mL and μM for each compound tested.**
(TIF)

## Acknowledgments

The authors thank M. Takagi for technical assistance and J. Akaki for helpful comments on the analysis of herbal plants. We also thank H. Nakayama, A. Kamo, H. Matsuda, and K. Ishii for technical advice.

## Author Contributions

**Formal analysis:** Yasuko Yoshioka, Yayoi Kamata.

**Funding acquisition:** Kenji Takamori.

**Investigation:** Yasuko Yoshioka, Yayoi Kamata, Yoshie Umehara, Ikuyo Yoshida.

**Methodology:** Yayoi Kamata, Mitsutoshi Tominaga.

**Project administration:** Yayoi Kamata, Mitsutoshi Tominaga.

**Supervision:** Kenji Takamori.

**Writing – original draft:** Yasuko Yoshioka, Mitsutoshi Tominaga, Nobuya Matsuoka.

**Writing – review & editing:** Yayoi Kamata, Mitsutoshi Tominaga, Kenji Takamori.

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
