## [Decision Letter · Decision Letter 0]

18 Nov 2020

PONE-D-20-28160

Extract of Scutellaria baicalensis induces semaphorin 3A production in human epidermal keratinocytes

PLOS ONE

Dear Dr. Takamori,

Thank you for submitting your manuscript to PLOS ONE. After careful consideration, we feel that it has merit but does not fully meet PLOS ONE’s publication criteria as it currently stands. Therefore, we invite you to submit a revised version of the manuscript that addresses the points raised during the review process.

Sorry for a delayed response as many potential reviewers are currently busy or not available to review this manuscript. Now your manuscript has been reviewed by three independent reviewers and they recommended the publication of your manuscript after a minor revision. Please address the comments from reviewer 2 and 3. Some extra experiments may be needed to address some of the comments.

We look forward to receiving your revised manuscript.

Kind regards,

Yi Cao

Academic Editor

PLOS ONE

Journal Requirements:

We note that one or more of the authors are employed by a commercial company: Kobayashi Pharmaceutical Co. Ltd..

2.1. Please provide an amended Funding Statement declaring this commercial affiliation, as well as a statement regarding the Role of Funders in your study. If the funding organization did not play a role in the study design, data collection and analysis, decision to publish, or preparation of the manuscript and only provided financial support in the form of authors' salaries and/or research materials, please review your statements relating to the author contributions, and ensure you have specifically and accurately indicated the role(s) that these authors had in your study. You can update author roles in the Author Contributions section of the online submission form.

2.2. Please also provide an updated Competing Interests Statement declaring this commercial affiliation along with any other relevant declarations relating to employment, consultancy, patents, products in development, or marketed products, etc.  

Reviewers' comments:

Reviewer's Responses to Questions

**Comments to the Author**

1. Is the manuscript technically sound, and do the data support the conclusions?

Reviewer #1: Yes

Reviewer #2: Partly

Reviewer #3: Yes

2. Has the statistical analysis been performed appropriately and rigorously? 

Reviewer #1: Yes

Reviewer #2: Yes

Reviewer #3: I Don't Know

3. Have the authors made all data underlying the findings in their manuscript fully available?

Reviewer #1: Yes

Reviewer #2: Yes

Reviewer #3: Yes

4. Is the manuscript presented in an intelligible fashion and written in standard English?

Reviewer #1: Yes

Reviewer #2: Yes

Reviewer #3: Yes

5. Review Comments to the Author

Reviewer #1: This is an interesting article on the development of a topical solution from plant extracts that utilizes small molecules that can penetrate the skin barrier to induce the production of the nerve repulsion factor Sema3A. The article is well written and the experimental protocol as well as the statistical analysis are sound.

Reviewer #2: Some skin diseases such as atopic dermatitis (AD) and xerosis is mainly due to hyperinnervation in the epidermis. And Semaphorin 3A (Sema3A) is a nerve repulsion factor expressed in keratinocytes and it suppresses nerve fiber elongation in the epidermis. Therefore the paper of Yasuko and co-workers find that S.baicalensis extract and its compounds, baicalin and baicalein, may be promising candidates for improving histamine-resistant itch via the induction of Sema3A expression in epidermal keratinocytes. Thus, the study may provide a new therapeutic candidates for improving histamine-resistant itch caused by epidermal hyperinnervation.

As such, the matter is of interest, however the paper suffers for some limits:

1)In Fig 3, It is unconvincing that simply use ELISA to measure the Sema3A protein levels in the culture supernatants of NHEKs treated with five tested samples. Sema3A is not only the secreted protein, but also abundant in intracellular of NHEKs. It is recommended to use WB or immunofluorescence to observe the protein levels in different treatment groups.

2)In Fig 3, samples No. 2, and 13 can increase the expression levels of Sema3A in cultured NHEKs at 24 hours. Do these drugs have any effect on cell viability?

3)In Fig5a, the results showed that the luciferase activities in cultured NHEKs were increased dose dependently by the application of baicalin orbaicalein. Are there statistically significant differences as compared to the control ?

Once the above concerns are fully addressed, the manuscript could be accepted for publication in this journal.

Reviewer #3: PONE-D-20-28160

Extract of Scutellaria baicalensis induces semaphorin 3A production in human epidermal keratinocyte

Reviewer’s comment　

Authors are interested in an importance of epidermal Sema3A level to prevent sensory nerve fiber elongation leading to serious scratching in pruritic skin diseases such as AD and xerosis. This manuscript describes herbal plant extracts were tested for an in vitro Sema3A induction activity in normal human keratinocytes (NHEKs). The major findings were obtained from gene promoter assay by Sema3A promoter that authors characterized, qPCR of Sema3A mRNA, ELISA of secretory Sema3A protein, and human skin equivalent model. The authors showed results of the primary screening of gene promoter activity for 91 herbal plant extracts. In further screening by qPCR of mRNA, the remaining was five herbal extracts No.2, 7, 10, 13 and 76. Finally, S. baicalensis extract was selected as the best one in screened samples. In addition, baicalein and its glucuronide are proved to be main active component among 5 flavonoids included in S. baicalensis.

Although the data are interesting and authors are expected to progress this research in future, I have the following concerns regarding the experiment and interpretation of the data.

1. Page 2 line 22: “Previous studies” is “Our previous studies”.

2. Page 3 line 36-39: Do authors think whether or not the active extract-induced Sema3A may lead to keratinocyte differentiation or migration?

3. Page 5 line 74: Authors mentioned Sema3A ointment in Abstract. Please add the reference here.

4. Page 6 line 87: Sample preparation is mentioned more in detail in the text.

5. Page 8 line 105, Page 11 line 160-161: According to pGL3-Sema3A vector containing luciferase gene, the sentence “Cultured NHEKs were……with luciferase reporter gene constructs containing -1444 bp of the human Sema3A promoter” is shown in Page 11, but more specified promoter sequence is shown in the text.

6. Page 10 line 143-146: Human skin equivalent model is supported by a little bit of more explanation for the readers.

7. Page 11 line 153: Statistical numbers (n) of the experiments are shown in the figures and legends.

8. Page 17 line 240-250: Did authors quantitate baicalein and other components in S. baicalensis extract that authors tested?

9. Page 20 line 296: It is referred that S. baicalensis contains 10 % baicalin and 1% baicalein. Which is the more active component? Do Authors guess beta-glucuronidase degrades baicalin faster, and leads to bioactive component baicalein? If so, smooth beta-glucuronidase degradation must be shown by an enzymatic experiment of culturing of keratinocyte or incubating of cell lysates.　

10. Page 21 line 311-312: Authors reported that Sema3A mRNA expression in NHEKs is regulated by calcium. How is the possibility of activation due to contaminated calcium in the tested samples?

11. Fig. 5 and the legend: Chemicals have been compared at same molar concentration. Chemicals are shown in the figure at each of molar concentrations. In the legend, each name of chemicals is followed by both units of ug/mL and uM.

6. PLOS authors have the option to publish the peer review history of their article (what does this mean?). If published, this will include your full peer review and any attached files.

Reviewer #1: No

Reviewer #2: No

Reviewer #3: No

---

## [Author Response · Author response to Decision Letter 0]

2 Apr 2021

Reviewer 1: I have incorporated all of your suggestions into my revision. They were very helpful. Thank you.

Reviewer 2: I have incorporated all of your suggestions into my revision. They were very helpful. Thank you.

---

## [Decision Letter · Decision Letter 1]

12 Apr 2021

Extract of Scutellaria baicalensis induces semaphorin 3A production in human epidermal keratinocytes

PONE-D-20-28160R1

Dear Dr. Takamori,

We’re pleased to inform you that your manuscript has been judged scientifically suitable for publication and will be formally accepted for publication once it meets all outstanding technical requirements.

Kind regards,

Yi Cao

Academic Editor

PLOS ONE

Additional Editor Comments (optional):

Reviewers' comments:

Reviewer's Responses to Questions

**Comments to the Author**

1. If the authors have adequately addressed your comments raised in a previous round of review and you feel that this manuscript is now acceptable for publication, you may indicate that here to bypass the “Comments to the Author” section, enter your conflict of interest statement in the “Confidential to Editor” section, and submit your "Accept" recommendation.

Reviewer #3: All comments have been addressed

2. Is the manuscript technically sound, and do the data support the conclusions?

Reviewer #3: Yes

3. Has the statistical analysis been performed appropriately and rigorously? 

Reviewer #3: I Don't Know

4. Have the authors made all data underlying the findings in their manuscript fully available?

Reviewer #3: Yes

5. Is the manuscript presented in an intelligible fashion and written in standard English?

Reviewer #3: Yes

6. Review Comments to the Author

Reviewer #3: (No Response)

7. PLOS authors have the option to publish the peer review history of their article (what does this mean?). If published, this will include your full peer review and any attached files.

Reviewer #3: No

---

## [Editor Report · Acceptance letter]

16 Apr 2021

PONE-D-20-28160R1 

Extract of *Scutellaria baicalensis* induces semaphorin 3A production in human epidermal keratinocytes 

Dear Dr. Takamori:

I'm pleased to inform you that your manuscript has been deemed suitable for publication in PLOS ONE. Congratulations! Your manuscript is now with our production department. 

Kind regards, 

on behalf of

Dr. Yi Cao 

Academic Editor

PLOS ONE